# Improved Perceptron of Subsurface Chlorophyll Maxima by a Deep Neural Network: A Case Study with BGC-Argo Float Data in the Northwestern Pacific Ocean

Jianqiang Chen [1], Xun Gong [2,3], Xinyu Guo [4], Xiaogang Xing [5], Keyu Lu [3], Huiwang Gao [6,7] and Xiang Gong [1,*]

1   School of Mathematics and Physics, Qingdao University of Science and Technology, Qingdao 266061, China; angyi_jq@163.com
2   Hubei Key Laboratory of Marine Geological Resources, China University of Geosciences, Wuhan 430074, China; gongxun@cug.edu.cn
3   Shandong Provincial Key Laboratory of Computer Networks, Qilu University of Technology (Shandong Academy of Sciences), Jinan 250101, China; ucfalux@ucl.ac.uk
4   Center for Marine Environmental Study, Ehime University, 2-5 Bunkyo-cho, Matsuyama 790-8577, Japan; guoxinyu@sci.ehime-u.ac.jp
5   State Key Laboratory of Satellite Ocean Environment Dynamics, Second Institute of Oceanography, Ministry of Natural Resources, Hangzhou 310012, China; xing@sio.org.cn
6   Qingdao National Laboratory for Marine Science and Technology, Qingdao 266237, China; hwgao@ouc.edu.cn
7   Key Laboratory of Marine Environment and Ecology, Ministry of Education of China, Ocean University of China, Qingdao 266071, China
*   Correspondence: gongxiang@qust.edu.cn

**Abstract:** Subsurface chlorophyll maxima (SCMs), commonly occurring beneath the surface mixed layer in coastal seas and open oceans, account for main changes in depth-integrated primary production and hence significantly contribute to the global carbon cycle. To fill the gap of previous methods (in situ measurement, remote sensing, and the extrapolating function based on surface-ocean data) for obtaining SCM characteristics (intensity, depth, and thickness), we developed an improved deep neural network (IDNN) model using a Gaussian radial basis activation function to retrieve the vertical profile of chlorophyll $a$ concentration (Chl $a$) and associated SCM characteristics from surface-ocean data. The annually averaged SCM depth was further incorporated into the bias term and the Gaussian activation function to improve the estimation accuracy of the IDNN model. Based on the Biogeochemical-Argo (BGC-Argo) data acquired for three regions in the northwestern Pacific Ocean, vertical Chl $a$ profiles produced by our improved DNN model using sea surface Chl $a$ and sea surface temperature (SST) were in good agreement with the observations, especially in regions with low surface Chl $a$. Compared to other neural-network-based models with one hidden layer and a sigmoid activation function, the IDNN model retrieved vertical Chl $a$ profiles well in more eutrophic subpolar regions. Furthermore, the application of the IDNN model to infer vertical Chl $a$ profiles from remote-sensing information was validated in the northwestern Pacific Ocean.

**Keywords:** subsurface chlorophyll maximum; deep neural network; Gaussian radial basis activation function; BGC-Argo; northwestern Pacific Ocean

## 1. Introduction

The ocean plays an important role in the global carbon cycle, with marine phytoplankton accounting for ~50% of the global primary production [1]. In particular, the subsurface chlorophyll maximum (SCM) layer contributes up to 75% of the depth-integrated primary production in open oceans and coastal seas [2–4]. Moreover, the phytoplankton carbon content in the SCM layer is positively correlated with the downward flux of particulate organic carbon (POC) to the deep sea [5]. Over the period of a year, 59–73% of the ocean is

expected to have the SCM layers with vertical scales from a few meters to tens of meters, where chlorophyll a concentration (Chl *a*) presents a peak value [6–8]. Therefore, characterizing SCM (intensity, depth, and thickness) is important for understanding marine primary production, the carbon cycle, and ongoing global warming trends.

Since the 1950s, based on in situ measurements, SCM characteristics have been examined from vertical distributions of Chl *a* concentration, which are either measured in discrete samples typically taken at standard depths or estimated continuously using in vivo fluorometry [9–12]. Cullen [7] summarized the observation of SCM characteristics since the 1940s based on the benchmark studies by Riley and Steele [13–15]. Anderson [9] reported that the high values of Chl *a* measured by in vivo fluorometry are typically confined to a layer between 55 and 65 m, with a peak value at 60 m in the northeast Pacific Ocean. In comparison, a prominent SCM between 65 and 150 m was found during two cruises in the western Pacific Ocean [16]. Taking the above as examples, SCM characteristics show spatial heterogeneity. However, not all in situ measurements capture SCM characteristics continuously and they depend on cruise lines or float trajectories, thus limiting the knowledge of SCM complexity in global oceans.

Since the late 1980s, efforts have been made to statistically extrapolate SCM characteristics from surface-ocean data, which can be detected by satellite. Morel and Berthon applied a shifted Gaussian function to categorize ~4000 vertical profiles of pigments to define seven Chl *a* hierarchies in open oceans and examine the associated SCM characteristics [17,18]. Uitz [19] upgraded the shifted Gaussian function for two ocean conditions that represent vertically stratified and mixed cases. Focusing on coastal seas, Richardson et al. [20] established a generalised linear model as a function of surface Chl *a* concentrations and surface water temperatures to estimate SCM characteristics in the southern Benguela upwelling systems. Xiu et al. [21] applied a blue-to-green band ratio algorithm and successfully retrieved SCM intensity from remote-sensing reflectance in the Bohai Sea of China. Although the retrieving functions based on surface-ocean data can provide Chl *a* concentration data for vast and continuous ocean areas, they show limitations in characterizing the SCM complexity due to the inadequacy of statistical adaptability at finer temporal resolutions (seasonal, daily, and smaller spatio-temporal scales).

Recently, neural-network-based algorithms have been regularly used to calculate SCM from surface-ocean data [22–24]. An advantage of neural networks is that they provide a near-real-time depiction of the SCM characteristics from remote-sensing data. For instance, Sammartino et al. [24] trained an artificial neural network (ANN) model with one hidden layer (also known as a shallow ANN) to infer Chl *a* vertical profiles in the Mediterranean Sea via in situ measurements together with remote-sensing data (Chl *a* and temperature). The shallow ANN model works well in the regions with low surface Chl *a* concentrations, but its performance decreases for the area with high Chl *a* surface concentrations [23,24]. It is known that the number of hidden layers in neural-network-based algorithms are one important factor controlling the estimation accuracy [24–26]. More importantly, a neural-network-based algorithms' estimation accuracy depends on the the type of activation function [26,27]. Compared with the logistic (sigmoid) activation function, a radial basis activation function is likely to produce better solutions for nonlinear problems [26]. Generally speaking, the previous neural-network-based models work better in retrieving SCMs than the traditional statistics-based models did. However, because previous neural-network-based models used only one genu in the hidden layer with the sigmoid activation function, their accuracy needs to improve for the vertical Chl *a* profiles.

In this study, we develop a deep neural network (DNN) model to retrieve the vertical profiles of Chl *a* from surface-ocean data which are equivalent to the average value within the first 20 m water depth [4]. Our DNN model has at least three hidden layers in the algorithm. Compared to the shallow ANN, such an improvement theoretically helps to improve performance for prediction capability [28]. Rather than using the sigmoid activation function, we use a Gaussian radial basis activation function in the DNN model, which is one of the most frequently used radial functions in the literature [26]. We train the DNN

model by inputting sea surface temperature (SST) and surface Chl *a* from Biogeochemical-Argo (BGC-Argo) floats. We apply the DNN model to the northwestern Pacific Ocean and compare the estimated vertical Chl *a* profiles and associated SCM characteristics with observations in different regions and seasons. Finally, we examine the prediction capability of our DNN model in retrieving vertical Chl *a* profiles from remote-sensing data in the northwestern Pacific Ocean.

## 2. Data and Methods

### 2.1. Improved DNN Model

The DNN model is an extension of a conventional ANN, with at least two hidden layers between the input and output layers. Because each node in the hidden layer makes both associations and grades of the input to determine the output, stacking more of these layers upon each other benefits more from multiple hidden layers. Generally, the running of a DNN model includes two steps: error forward and error backward propagation. Forward propagation refers to the DNN receiving signals from the input layer and then passes the signals to the hidden layer (Figure 1). After processing the signals by the neurons in the hidden layer, the DNN passes them to the output layer. Then, by comparing and calculating the output error with the target, backward propagation is used to adjust the weight of signals in the hidden layers and further reduce the error using the optimization algorithm.

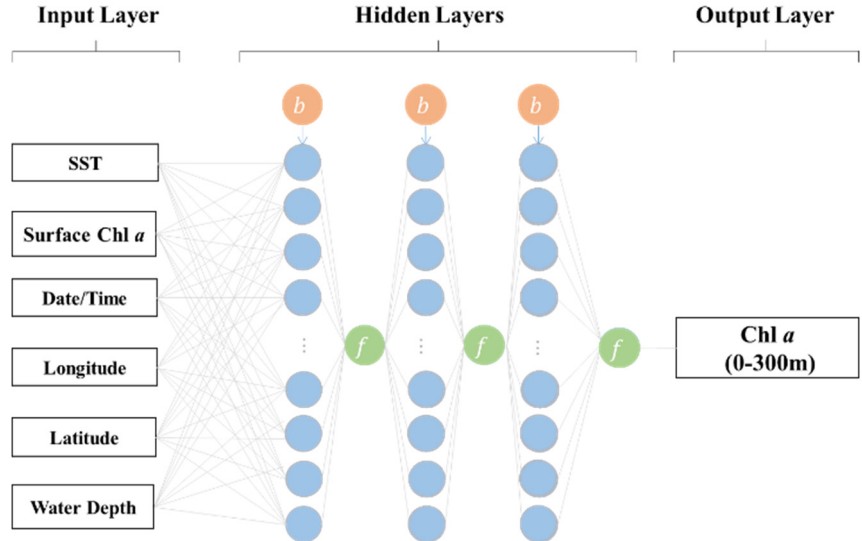

**Figure 1.** Structure of the deep neural network (DNN). The input elements of DNN are longitude, latitude, time, sea surface temperature (SST), sea surface Chl *a*, water depth. The output is the vertical distribution of Chl *a* concentration over the water depth of 0–300 m. *b* is the bias term in the hidden layer. The prior information of nonlinear activation function (*f*) connects the hidden layers to the output.

In the calculation of SCM characteristics, we improved the existing DNN model in the following two aspects. First, we improve the capability of model for the calculation of SCM depth by replaced the bias term *b* (that is, the intercept term) from random values to annual mean of SCM depths (Equation (1)).

$$b_{SCM} = \frac{avg(z_{\max})}{\max(z)},$$  (1)

where $z$ is the water depth, $\max(z)$ is set to 300 m by assuming that there is no Chl *a* below 300 m depth, and $avg(z_{\max})$ is the annual mean of SCM depth ($z_{\max}$).

In the second aspect, we improve the calculation accuracy of the model for the SCM intensity and thickness. Considering the unimodal chlorophyll profiles, a Gaussian radial

basis function is substituted for the sigmoid function as a nonlinear activation function (*f*, Equation (2)) in the original DNN model to amplify the signals within the SCM layer. The advantage of Gaussian radial basis activation function is that it is similar to quadratic function for the center values of input variables, while the sigmoid activation function is similar to linear function about the moderate inputs [26].

$$f = e^{-\pi(X_j - b_{SCM})^2},$$ (2)

where $X_j$ is the $X$ value of the $j$th output in the hidden layer, and $b_{SCM}$ is the bias term computed by the annual mean of SCM depth (Equation (1)). The Gaussian radial basis activation function helps to absorb the information of the annually averaged SCM depth and, hence, further extract the SCM features.

Consequently, a DNN model with at least two hidden layers was applied due to its availability in capturing the nonlinear relationships, and the annual averaged SCM depth is incorporated into both the bias term and the Gaussian radial basis activation function to improve the capability in retrieving the SCM characteristics. We name this improved model the IDNN model.

### 2.2. BGC-Argo Data for the IDNN Model

The in situ data for the IDNN model were collected from 16 BGC-Argo profiling floats in the northwestern Pacific Ocean (https://biogeochemical-argo.org/, accessed on 19 April 2021). Figure 2 plotted the trajectories of the 14 BGC-Argo profiling floats within 123–180°E, 12–48°N, where a SCM feature was observed. Figure A1 showed the locations of vertical Chl *a* profiles observed from 16 BGC-Argo floats in the absence of a SCM. The acquired 2409 vertical Chl *a* profiles, covering four seasons during the period from July 2017 to April 2021, were used in our study after quality control to remove aberrant data caused by electronic noise [29].

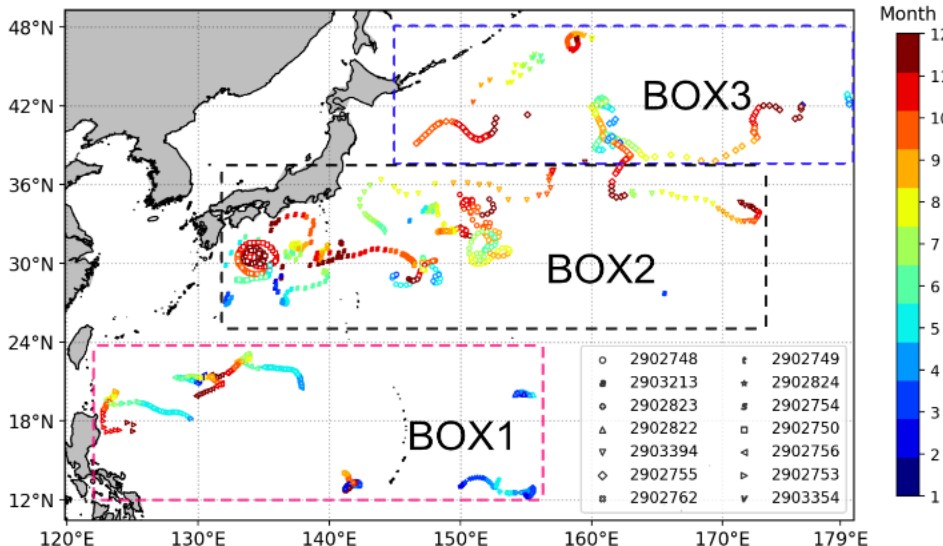

**Figure 2.** Locations and measuring months of 14 BGC-Argo profiles with a subsurface chlorophyll maximum (SCM) feature in the northwestern Pacific Ocean.

Most vertical profiles of Chl *a* showed a unimodal distribution, while the remaining minors showed either increasing or decreasing Chl *a* with depth. Here, focusing on the SCM patterns, we structure each BGC-Argo vertical profile based on a Gaussian function assumption [30] (Equation (3)).

$$Chl(z) = Ae^{-\frac{(z-z_{\max})^2}{2\sigma^2}}$$ (3)

where $\sigma$ is its standard deviation, $A$ is the amplitude of the Gaussian curve, and $z_{max}$ is the location of the amplitude. To quantify the vertical scale of the SCM layer, $2\sigma$ was used to represent the SCM thickness [2,31]. Because the upper layer of the SCM ($z_{max} - \sigma$) must be inside the water, it is set as a nonnegative value. That is, if $z_{max} - \sigma < 0$, the upper layer of the SCM is set to the sea surface (0 m). In addition, the SCM intensity refers to the peak value of Chl $a$ concentration ($A$).

The values of the Gaussian parameters ($\sigma$, $A$, and $z_{max}$) were obtained by fitting all observed vertical Chl $a$ profiles (Figure 3), which can be used to filter out Chl $a$ profiles with no significant SCM characteristics via the following three steps. First, profiles with the values of parameter $\sigma$ ranging from the lower limit of the data value to half of the upper limit (0–48 m) (Figure 3a), are kept, thereby leaving 1676 profiles. Second, parameter $A$ (the peak values of Chl $a$ obtained from Gaussian fitting) was assumed to be at least twice the surface Chl $a$ concentrations. Meanwhile, values larger than the upper limit (2.2 mg m$^{-3}$) were neglected as outliers (Figure 3b). This step excluded 328 profiles. Third, $z_{max}$ values are limited to depths above 200 m. Finally, after visually reviewing all the filtered profiles, 1342 out of the total 2409 profiles were retained in the following analysis. Consequently, the remaining vertical Chl $a$ profiles present significant SCM characteristics.

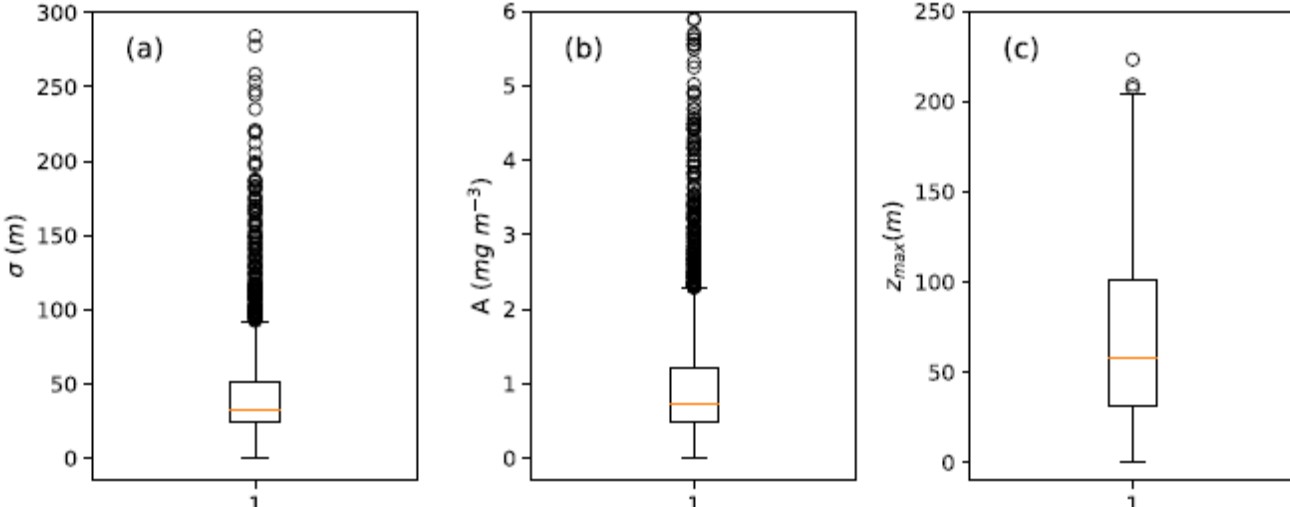

**Figure 3.** Boxplot of three parameters (standard deviation (**a**); amplitude of the Gaussian curve (**b**); location of the amplitude (**c**)) in the Gaussian function in the northwestern Pacific Ocean. In each box, the orange horizontal line represents the median. The upper and lower horizontal lines of the box represent the 75th and 25th percentiles (Q3 and Q1), respectively. The upper and lower horizontal whiskers lines are the upper limit (U) and lower limit (L) of the data, respectively; the circles represent the outliers.

The remaining 1342 profiles with SCM covered four seasons from February 2018 to April 2021 (Table 1, Figure 2), while those profiles that are excluded in the initial quality control are mostly characterized by the absence of SCM during the winter season (Figure A1). In general, vertical Chl $a$ profiles with a SCM accompany higher SSTs and lower surface Chl $a$ concentrations in subtropical and subpolar areas, compared to the profiles without SCM. For example, in subtropical area (BOX2 in Figure 2), for the profiles with SCM, the average surface Chl $a$ concentration is about 0.14 mg m$^{-3}$, and the average SST is about 24.3 °C; for the profiles without SCM, the average surface Chl $a$ and SST is 0.72 mg m$^{-3}$ and 19.6 °C, respectively. In tropical area (BOX1 in Figure 2), no significant difference between profiles with and without SCM was found from averaged surface Chl $a$ concentration and SST.

**Table 1.** Averaged intensity and depth of SCM in BGC-Argo database used (Year/Month/Day).

| Region | Float No. (Number of Profiles) | Data Duration | SCM Intensity (mg m$^{-3}$) | SCM Depth (m) |
|---|---|---|---|---|
| BOX1 (12–24°N, 123–156°E) | 2902753 (118) | 2019/3/30–2019/12/8 | 0.57 (0.37–0.85) | 117 (88–166) |
| | 2902756 (184) | 2019/3/25–2020/12/2 | 0.67 (0.34–0.73) | 115 (88–150) |
| | 2902762 (82) | 2020/8/16–2021/4/18 | 0.43 (0.29–0.77) | 139 (90–175) |
| | 2902822 (37) | 2021/1/12–2021/4/17 | 0.43 (0.29–0.53) | 127 (97–149) |
| | 2902823 (30) | 2021/1/17–2021/4/17 | 0.41 (0.19–0.55) | 147 (129–183) |
| | 2902824 (30) | 2021/1/20–2021/4/16 | 0.47 (0.39–0.55) | 153 (127–176) |
| Seasonal average (Winter, Spring, Summer, Autumn) | | | 0.44, 0.59, 0.58, 0.57 | 134, 124, 109, 128 |
| BOX2 (26–38°N, 132–173°E) | 2902748 (199) | 2018/5/31–2021/4/17 | 1.23 (0.55–3.54) | 76 (30–117) |
| | 2902749 (28) | 2018/5/31–2018/9/8 | 1.07 (0.61–2.18) | 79 (40–108) |
| | 2902750 (108) | 2018/9/13–2019/5/31 | 0.85 (0.44–1.67) | 89 (29–118) |
| | 2902754 (147) | 2018/8/30–2021/4/16 | 1.1 (0.32–6.65) | 76 (23–135) |
| | 2902755 (9) | 2019/10/19–2019/11/28 | 0.65 (0.58–1.17) | 40 (18–62) |
| | 2903213 (1) | 2018/2/22–2018/2/22 | 0.91 (0.91–0.91) | 68 (68–68) |
| | 2903394 (78) | 2019/5/26–2020/12/8 | 0.79 (0.45–6.12) | 65 (28–100) |
| Seasonal average (Winter, Spring, Summer, Autumn) | | | 0.62, 1.18, 1.49, 0.91 | 74, 69, 74, 83 |
| BOX3 (38–48°N, 145–180°E) | 2902755 (204) | 2018/9/3–2021/4/16 | 1.92 (0.50–5.70) | 41 (12–96) |
| | 2903354 (87) | 2018/7/25–2019/9/4 | 2.5 (0.24–7.07) | 28 (6–49) |
| Seasonal average (Winter, Spring, Summer, Autumn) | | | 1.09, 1.60, 2.40, 2.00 | 59, 39, 33, 40 |
| Total area | (1342) | 2018/2/22–2021/4/18 | 1.14 (0.20–7.07) | 85 (6–183) |
| Seasonal average (Winter, Spring, Summer, Autumn) | | | 0.52, 0.88, 1.57, 1.17 | 115, 101, 72, 75 |

The average value of SCM depth and its intensity in each BGC-Argo float is shown in Table 1. The averaged SCM intensity from profiles of each BGC-Argo float in tropical area (12–24°N) has ranges between 0.41–0.67 mg m$^{-3}$ over depths of 115–153 m; the averaged SCM intensity in subtropical (26°N–38°N) is about 0.65–1.2 mg m$^{-3}$ with shallower SCM depths over 40–89 m. At high latitudes (38–48°N), the SCM exists at a depth <50 m with the largest intensity larger than 1.5 mg m$^{-3}$. Table 1 also presents the seasonal averaged SCM depth and intensity. Compared with summer and autumn, the SCMs get deeper and weaker in winter and spring, which is probably due to increased vertical mixing in winter and spring. This indicates that physical entrainment may extract some of phytoplankton from the SCM layer to the surface layer and thereby reduce the SCM intensity [2,32]. Generally, the SCM characteristics differ among the three regions that range from tropical area to subtropical area and then to subpolar area, with a weaker seasonal variation. Therefore, based on spatial variations of SCM characteristics in the northwestern Pacific Ocean, the remained 1342 vertical profiles of Chl *a* are classified into three BOXes (Table 1). In BOX1, 6 BGC-Argo floats contained 481 profiles span in the tropical Pacific. 570 profiles from 7 BGC-Argo floats near 30°N belong to BOX2 in subtropical area, and the remaining 291 profiles from 2 BGC-Argo floats locate at BOX3 in subpolar Pacific.

### 2.3. Satellite Data for the IDNN Model

To evaluate the performance of IDNN reconstruction using remote-sensing data, the MODIS Level 3 standard mapped image monthly Chl a and SST database with a 9 km spatial resolution were downloaded from Asia-Pacific Data-Research Center (ARDRC, http://apdrc.soest.hawaii.edu/dods/public_data/satellite_product/MODIS_Aqua/, accessed on 19 April 2021) and used to extract the input values for the IDNN model.

Each profile of the BGC-Argo database remained above was then matched up with satellite data of surface Chl a and SST using the bilinear interpolation method [33]. This matchup process led to keeping a number of 985 BGC-Argo profiles from 2017 to 2021. The matchup file includes both training and test points.

Before the training, SST and Chl a data from BGC-Argo profiles and satellite were standardized to make them dimensionless and have the same order of magnitude. Thus, for each point, all data are clustered around 0 with a variance of 1.

### 2.4. Training Process

In our IDNN model, SST, sea surface Chl *a*, associated geo-location (latitude, longitude) and observation time, and water depth were selected as input variables (Figure 1), which are similar to the study by Sammartino et al. [24]. The reason for choosing geo-location and observation time is that the SCM characteristics varied with seasonality and spatiality (Table 1). SST is related with vertical stratification of the water column, influencing SCM characteristics. Moreover, the choice is supported by the potential capability of the network to find the nonlinear relationship between the input variables and Chl *a* concentration at different depths of the water column.

The IDNN model was trained in each BOX, respectively. Seventy-five percent of the input data and vertical profiles of Chl *a* from BGC-Argo floats in each BOX (Table 1) were segmented and fed into the IDNN model as the training set, which was selected randomly. We conducted 10 experiments by selecting 75% of the input data randomly in each BOX. The RMSEs ranged between 0.0033–0.0038 mg m$^{-3}$ in BOX1, 0.036–0.044 mg m$^{-3}$ in BOX2, and 0.25–0.36 mg m$^{-3}$ in BOX3. The results indicate the stability by randomly selecting training data. Gradient descent was applied to reverse propagation of the IDNN model, which was trained using an Adam optimizer. Moreover, to avoid over-fitting, we applied the dropout technique to discard some neurons. During the training process, 15% of the training set was randomly selected as a validation set to verify whether the IDNN model was over-fitted. According to the performance of the validation set, the parameters of the IDNN model were determined using a grid search. Overall, after stable network training, three IDNN model frames were determined for each BOX region (Table 2). Based on the best calculation performance, the parameters in BOX1 and BOX2 are equivalent to each other, but differ from those in BOX3.

**Table 2.** Parameters of the IDNN model in three BOXes.

| Network Parameter | Parameters | | |
|:---:|:---:|:---:|:---:|
| | **BOX1** | **BOX2** | **BOX3** |
| Hidden layer depth | 3 | 3 | 4 |
| Number of hidden neurons | 64, 64, 64 | 64, 64, 64 | 64, 128, 128, 64 |
| Momentum | 0.9 | 0.9 | 0.9 |
| Epoch | 115 | 115 | 150 |
| Learning rate | 0.01 | 0.01 | 0.01 |
| Dropout rate | 0.1 | 0.1 | 0.1 |

To evaluate the errors between the IDNN output and the observed value, the determination coefficient ($R^2$), correlation coefficient ($\rho$), root mean square error (RMSE), mean absolute percentage error (MAPE), and mean bias error (MBE) are introduced as evaluation indicators. Their formulas are shown in Table A1. The $R^2$ and RMSE values were used as performance indicators to evaluate the effectiveness of the developed models. MAPE and MBE capture the average difference between the estimated and observed values.

### 3. Results and Discussion

#### 3.1. IDNN-Retrieved Vertical Chl a Profiles

After evaluating the IDNN performance on a training set, we applied the IDNN model to a test set. Here, the test set contains 25% of the total BGC-Argo surface Chl *a* and SST datasets. In the testing phase of the IDNN model, a trained network was used for forward estimation. The data on the test set are normalized in a similar manner to that in the training set, with the network output being inversely normalized to the original unit.

As shown in Figure 4a–c, the modeled Chl *a* concentrations from the test set are closely comparable with the observed values for the upper 300 m in each BOX. Most of the data are close to the bisector (the black dashed line in Figure 4a–c), especially for Chl *a* concentration of ~0.1 mg m$^{-3}$. Owing to the high noise/signal ratio, the low Chl *a* concentration shows a relatively higher offset [25]. Compared with the BGC-Argo profiles of Chl *a* in each BOX, our IDNN modeling results show smaller fluctuations in the surface and SCM layers, except for the place close to the SCM depth in the BOX3 where the standard variances of the model results are larger than those from observations (Figure 4d–f). The variability in the shapes of vertical Chl *a* profiles between 0–50 meters in the in situ observations of BOX2 (green shade in Figure 4e) is mainly caused by two out of 16 profiles, which present the SCM depth above 50 m with the intensity over 4 mg m$^{-3}$. After removing these two profiles, the green shaded regions between 0–50 meters in BOX2 show smaller fluctuations (Figure A2). This result suggests that, even with some episodic signals, the network model has good robustness and can infer the shapes of vertical Chl *a* profiles from surface data only.

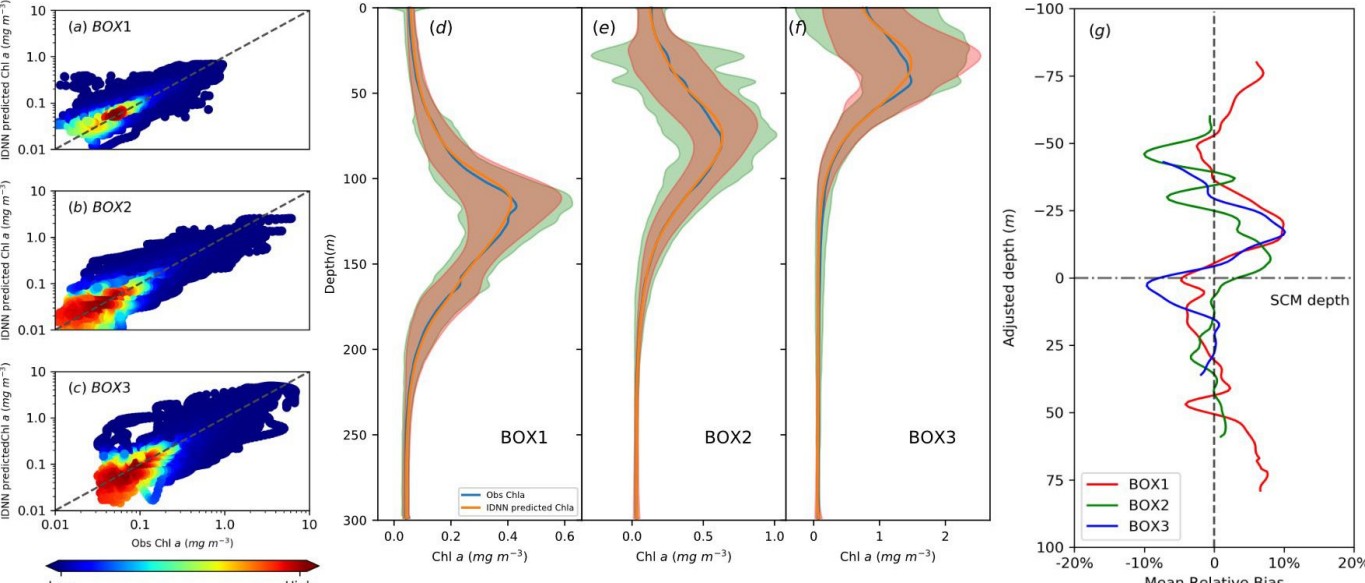

**Figure 4.** (**a–c**) Scatter plot of observed Chl *a* concentration (*x*-axis) and estimated Chl *a* value (*y*-axis) in BGC-Argo test set in BOXes 1–3. The black dashed line is the bisector of the first quadrant, i.e., y = x. (**d–f**) The mean of observed value (blue line) and the mean of IDNN predicted values (orange line) in each BOX. The pink and green shades are the standard variance of the model results and observations, respectively, which overlap and form the brown shade. (**g**) The mean relative bias as a function of the adjusted depth between observed and the IDNN predicted Chl *a* in the test set for the three BOXes. The adjusted depth is defined as the difference between the observed SCM depth and the modeled one.

Statistically, the relative biases show that maximal bias (8–10%) occurs at about 10–20 m shallower than the SCM depth in BOX1 and BOX2, while in BOX3 there are large relative biases (about ±10%) at the SCM depth and about 20 m above (Figure 4g). The relative biases in BOX1 and BOX2 are retrieved by smaller values compared to those in BOX3, indicating that the performance of the IDNN model is better for BOX1 and BOX2.

The statistical indices calculated for the IDNN assessment from the test set are listed in Table 3. For all three BOXes, high R$^2$ values suggest that input variables (the surface Chl *a*, SST, acquisition date and geo-location) can explain over 71% of Chl *a* variability at different depths. Here, the calculated R$^2$ values are consistent with those reported by Sammartino et al. [24], owing to similar input variables. Moreover, the high correlation coefficient ($\rho > 0.87$) suggests the goodness of IDNN model performance, with a slight underestimation of the observed Chl *a* in BOX3. Statistically, RMSEs less than 0.11 mg m$^{-3}$

and MAPEs from 0.036 to 0.13 in three BOXes reveal a small divergence between the predicted values and observed values. The statistical results indicate a robust prediction capability of the IDNN in three BOXes.

**Table 3.** Statistical results of the comparison between the observed Chl *a* values and the predicted Chl *a* concentration by the IDNN models using BGC-Argo data. $R^2$ refers to the Determination Coefficient, $\rho$ to the Pearson's correlation coefficient, RMSE denotes the root mean square error, and MAPE represent the mean absolute percentage error.

| Index | Region | | |
|:---:|:---:|:---:|:---:|
| | BOX1 | BOX2 | BOX3 |
| $R^2$ | 0.77 | 0.72 | 0.71 |
| $\rho$ | 0.89 | 0.88 | 0.87 |
| RMSE | 0.0040 | 0.025 | 0.11 |
| MAPE | 0.036 | 0.073 | 0.13 |

*3.2. IDNN-Retrieved SCM Characteristics*

We apply the IDNN model to calculate the seasonality of SCM characteristics. As shown in Figure 5, the regionally averaged Chl *a* profile predicted by the IDNN model from surface data is comparable to the observations for four seasons. In general, the predicted Chl *a* concentrations fluctuated within the observation variances. Moreover, the IDNN retrieves the vertical Chl *a* profile with better performance for the area with low surface Chl *a* values than the area with high surface Chl *a* values. For instance, the IDNN presents the best estimation for spring to autumn in BOX1 (Figure 5c). In Figure 5f–g, large standard variances of observed profiles occurred between 0–50 m (green shades), which is similar to those in Figure 4e. Therefore, we removed the two profiles with an episodically stronger SCM within 0–50 m in BOX2 during summer and autumn, and then the two variance spikes in 0–50 m disappear (Figure A2). These results highlight the capability of a deep neural algorithm to extend the surface-ocean information to SCM layer with seasonal variation. For the non-standard bell-shape of vertical Chl a profiles with higher surface Chl a values (Figure 5h–k), the IDNN predictions close to SCM depths are slightly overestimated or underestimated with respect to the observed values, especially in BOX2 Winter and in BOX3 Spring. The high variance in surface Chl *a* (e.g., the green shaded area in Figure 5h–k) is related to dissimilar shapes of vertical Chl *a* profiles, and thus reduce the network prediction accuracy on SCM characteristics. However, beyond the SCM layer, the two lines (red and blue) coincide during four seasons, especially in summer and autumn (Figure 5h–k). Here, we attribute the relatively high estimation offset for the area with high surface Chl *a* values to a non-standard bell-shape of vertical Chl *a* profiles and seasonal inconsistency.

As shown in Table 4, the seasonally averaged SCM characteristics from the test set agree well with the observations in each BOX. The SCM thickness can be estimated by fitting vertical Chl *a* profiles using the generalized Gaussian function (Equation (3)). Since recent studies have developed the curve fitting of vertical Chl *a* profiles by superimposing the generalized Gaussian function onto a linearly [19] or exponentially decreasing background Chl *a* concentration [34], we compared the curve-fitting performance of the three approaches in three BOXes. The results show that the generalized Gaussian function has a higher goodness of fit than other two functions. For example, for the non-standard bell-shape of vertical Chl *a* profiles in BOX3 (Figure 5i–k), the generalized Gaussian function has the goodness of fit of 90%, following by the function combined an exponentially decreasing background (88%) and the function combined a linearly decreasing background (about 76%). Based on our statistical results, we estimate the SCM thickness by using the generalized Gaussian function (Equation (3)).

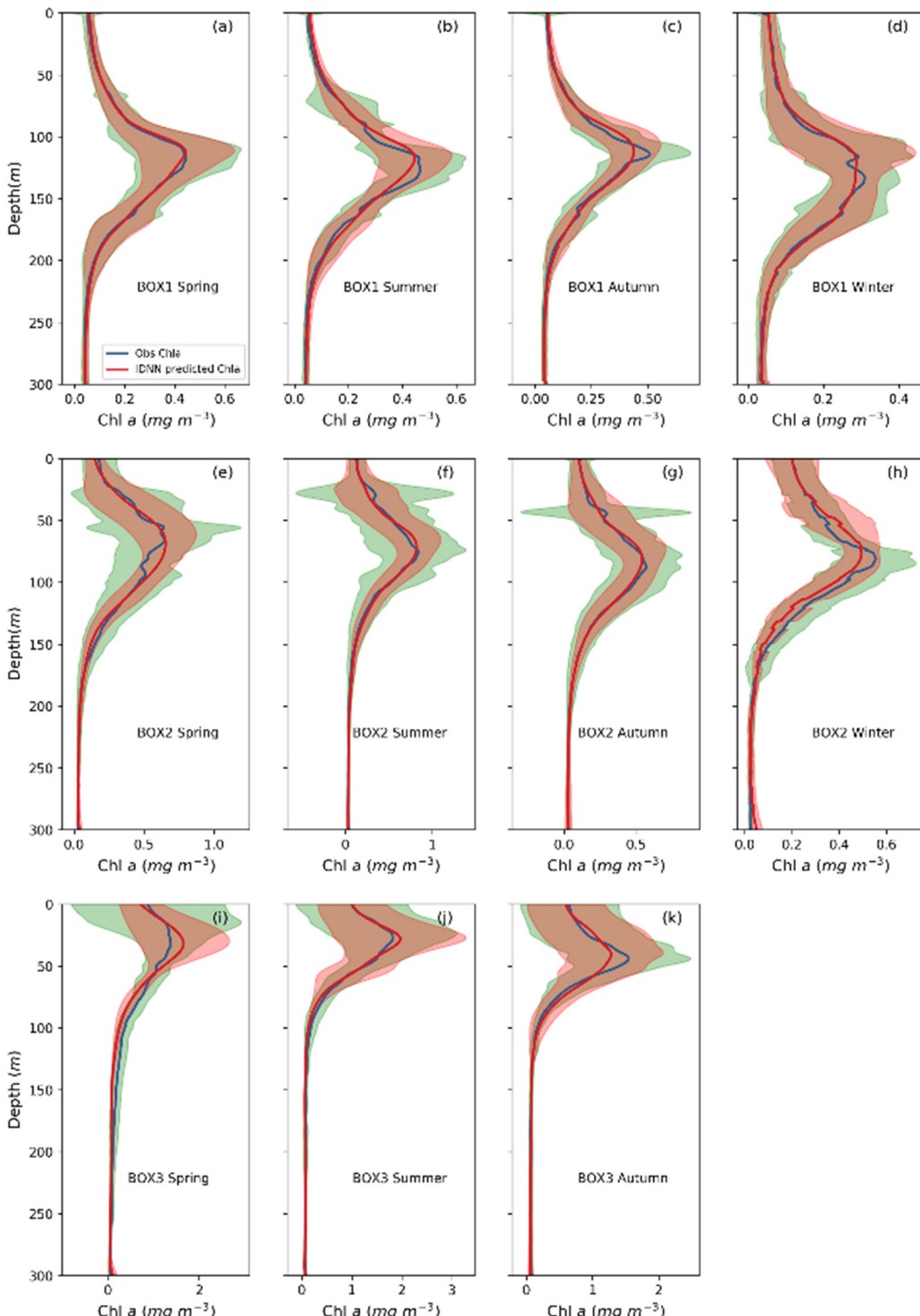

**Figure 5.** Aggregated chlorophyll vertical profiles from the test set in terms of seasons in BOX1 (**a–d**), in BOX2 (**e–h**), and in BOX3 (**i–k**). The blue and red solid lines represent the mean of observed value and the mean of IDNN predicted Chl *a*, respectively. The pink and green shades are the standard variance of the model results and observations, respectively, which overlap and form the brown shade.

Generally, MAPEs of all three SCM parameters are less than 35%, with an average value of 21%; MAPE of the SCM depth was the smallest (18% on average), which was followed by that of SCM intensity and thickness (Table 4). In addition, the averaged absolute relative bias is small in BOX1 (18%) and BOX2 (22%) but slightly larger in BOX3

(25%) due to the differences in SCM intensity during spring and SCM thickness in summer. Overall, these results indicate that our model captures the SCM characteristics well.

**Table 4.** Comparison of seasonal averaged SCM characteristics of three BOXes from test set with observations. Winter is defined as from December to February, and so on.

| Region | Season | SCM Depth (m) | | | SCM Intensity (mg m$^{-3}$) | | | SCM Thickness (m) | | |
|--------|--------|------|-------|------|------|-------|------|------|-------|------|
| | | Obs. | Model | MAPE | Obs. | Model | MAPE | Obs. | Model | MAPE |
| BOX1 | Winter | 136 | 136 | 9% | 0.45 | 0.37 | 20% | 70 | 73 | 21% |
| | Spring | 126 | 126 | 10% | 0.59 | 0.52 | 19% | 75 | 85 | 25% |
| | Summer | 120 | 125 | 19% | 0.62 | 0.50 | 21% | 75 | 78 | 31% |
| | Autumn | 110 | 115 | 8% | 0.58 | 0.47 | 19% | 75 | 80 | 15% |
| BOX2 | Winter | 76 | 75 | 14% | 0.62 | 0.54 | 21% | 80 | 76 | 21% |
| | Spring | 69 | 73 | 25% | 1.04 | 0.78 | 21% | 65 | 70 | 32% |
| | Summer | 71 | 75 | 16% | 1.13 | 1.01 | 28% | 43 | 56 | 22% |
| | Autumn | 80 | 82 | 12% | 0.94 | 0.65 | 27% | 54 | 64 | 26% |
| BOX3 | Spring | 40 | 35 | 35% | 1.80 | 1.70 | 30% | 69 | 56 | 20% |
| | Summer | 34 | 32 | 21% | 2.48 | 2.49 | 29% | 43 | 42 | 31% |
| | Autumn | 42 | 44 | 24% | 2.12 | 1.80 | 15% | 39 | 41 | 22% |

In addition to the seasonally averaged SCM characteristics in a region, our IDNN also successfully retrieved pixel-by-pixel variations along with the trajectory of a BGC-Argo float (Figure 6). The reconstructions of SCM characteristics in the test set show high consistency with the in situ observations along the BGC-Argo float No. 2902756 in BOX1, except for the SCM depth upwards during July and August 2020 (Figure 6a). In BOX2, the modeled SCM depth and thickness agree with observations along BGC-Argo float No. 2902748, while the SCM intensity was slightly overestimated during summer in 2018 and 2019 (Figure 6b). Similar to BOX2, the IDNN estimation shows relatively higher summer SCM intensities in BOX3 (e.g., BGC-Argo float No. 2902755) compared to the observations (Figure 6c).

*3.3. Role of the Gaussian Activation Function in Enhancing Estimation Accuracy*

As explained in Section 2.1, our IDNN model has two improvements compared to previous DNN model: the bias term and the Gaussian activation function. To anatomically determine the contribution of each modification in our IDNN model, we conducted two additional experiments with each stand-alone improvement.

(i)   DNN model using a sigmoid activation function with bias improvement by incorporating SCM depth (Equation (1)) (hereafter, referred to as DNN-b);

(ii)  DNN model using random bias values with a Gaussian activation function, rather than a sigmoid function (Equation (4)) (hereafter, referred to as DNN-G).

$$f = e^{-\pi(X_j - b)^2}; \tag{4}$$

where $X_j$ is the $X$ value of the $j$th output in the hidden layer, $b$ is the bias term with random values.

As shown in Figure 7, the DNN-b failed to reproduce the SCM characteristics, especially in BOX2. Compared to the observations, the SCM intensities predicted by the DNN-b in BOX1 and BOX3 were smaller, and the SCM depth in BOX3 was shallower. The DNN-G applied the Gaussian activation function to retrieve the SCM characteristics successfully, although the estimated SCM depth in BOX3 was slightly shallower than the observed one (Figure 7). Thus, we deduce that the application of the Gaussian activation function, rather than the sigmoid function, significantly improves the capability of the DNN model in estimating SCM characteristics.

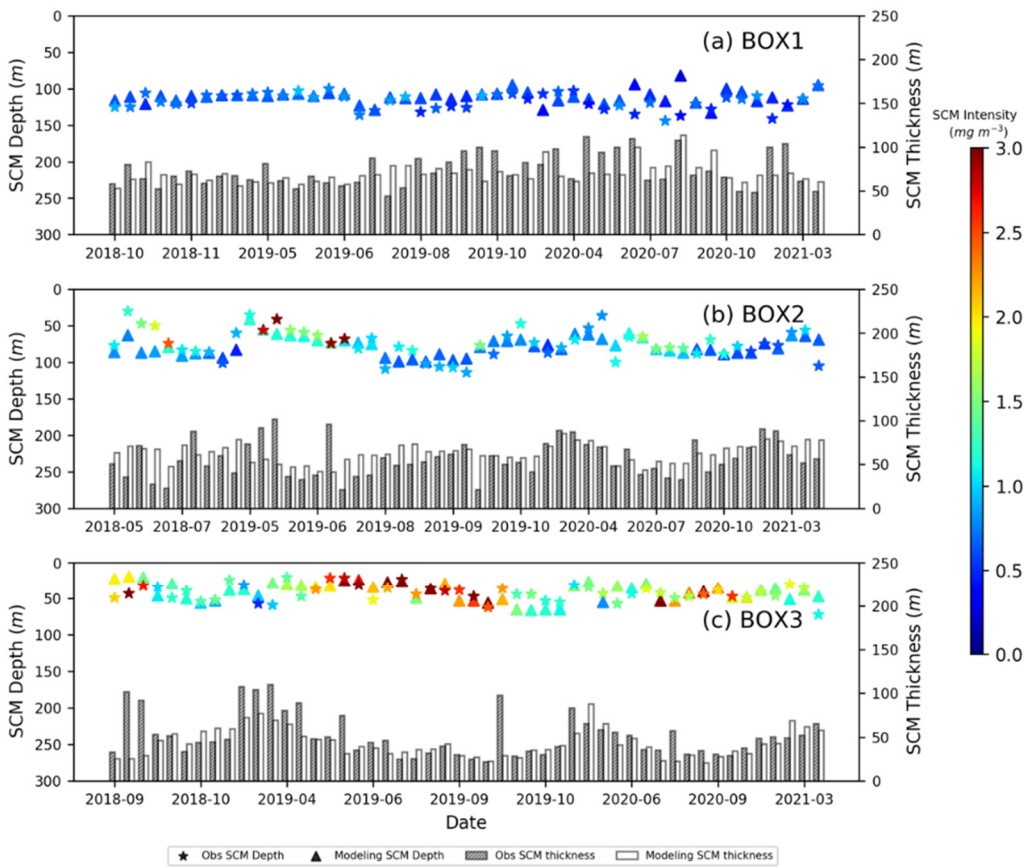

**Figure 6.** Comparison of SCM characteristics (depth, intensity, and thickness) obtained from the IDNN results in the test set and observation profiles along trajectories of three BGC-Argo (No. 2902756 (**a**), No. 2902748 (**b**), and No. 2902755 (**c**) in BOX1–BOX3, respectively).

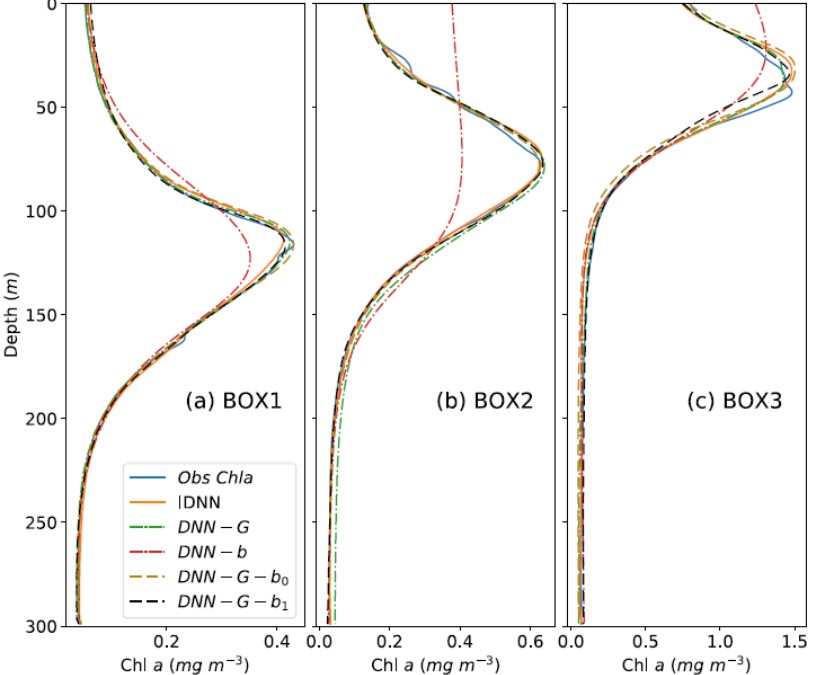

**Figure 7.** Aggregated chlorophyll vertical profiles from the test set in different experiments in BOXes 1–3 (**a**–**c**).

Furthermore, towards BOX3 in the subpolar areas with high surface Chl *a* concentration, we conducted two other experiments to explore the role of the Gaussian activation function in learning the SCM characteristics in based on:

(iii)　DNN-G model in which the bias term *b* was set as 0 (Equation (5)) (hereafter, referred to as DNN-G-b$_0$),

$$f = e^{-\pi X_j^2};\tag{5}$$

(iv)　DNN-G model in which the bias term *b* was set as 1 (Equation (6)) (hereafter, referred to as DNN-G-b$_1$),

$$f = e^{-\pi (X_j-1)^2}.\tag{6}$$

The predicted chlorophyll values in DNN-G-b$_0$ and DNN-G-b$_1$ were similar to the IDNN results, especially for BOX1 and BOX2 (Figure 7a,b). In BOX3, the SCM intensity captured by these two experiments varied slightly from the values of bias, while the SCM depths remained generally constant (Figure 7c). Our results indicate that the improvement in model learning using different bias values based on the Gaussian activation function is gentle.

### 3.4. Comparison with Shallow ANNs

Studies have used shallow ANN models to estimate the vertical distribution of Chl *a* in open oceans and coastal seas [22–24]. The shallow ANN model is a multi-layer perceptron (MLP), which consists of input layers, output layers, and one hidden layer between them. In the northwestern Pacific Ocean around Japan, Osawa et al. [22] applied a shallow ANN model with six input variables (SST, surface Chl *a*, mixed layer depth, geo-location, and Julian days), 50 nodes in the hidden layer to output four Gaussian parameters for estimating vertical profiles of Chl *a*. In the Mediterranean Sea, Sammartino et al. [24] reconstructed vertical profiles of Chl *a* by employing a shallow ANN model in which input variables included SST, surface Chl *a*, geo-location, and day of the year. The shallow ANN model works well in the regions with low surface Chl *a* concentrations but its performance decreases for the area with high Chl *a* surface concentrations [23,24].

To test the advantages of including more hidden layers, we perform calculations by two ANN models with one hidden layer (MLP-1 thereafter) and three hidden layers (MLP-3 thereafter), respectively. Both MLP models are input the same data. Moreover, the MLP models were trained using the same training and test sets that were employed to train the IDNN model. The comparison was carried out on the test sets.

The relative deviation between MLP models and the IDNN model were plotted side by side in Figure 8. Compared to our IDNN model results, the mean value of R$^2$ and $\rho$ obtained by the MLP-1 and MLP-3 decrease in all three BOXes, while RMSE and MAPD increase significantly in BOX2 and BOX3 (>55%), especially obtained by the MLP-1 model (86–196%). This suggests that the IDNN model exhibited a better performance, and the MLP-3 model has a similar capability with the IDNN model in the tropical sea area (BOX1).

In terms of SCM intensity and thickness, the MLP-3 model performed better than the MLP-1, although the mean values were underestimated by the MLP models in BOX1 and BOX2 (Figure 9a,b). While in the BOX3, both the MLP-1 and MLP-3 models failed to reveal the presence of SCM in the vertical Chl *a* profiles (Figure 9c). Therefore, the MLP-3 model is more capable of predicting SCM characteristics than the MLP-1 model; however, both MLP models are not adapted to the subpolar sea areas.

### 3.5. Application of the IDNN Model to Satellite Data

To evaluate the IDNN prediction capability using remote-sensing data, the Chl *a* concentrations retrieved by the IDNN model were compared with BGC-Argo observations at the matchup stations.

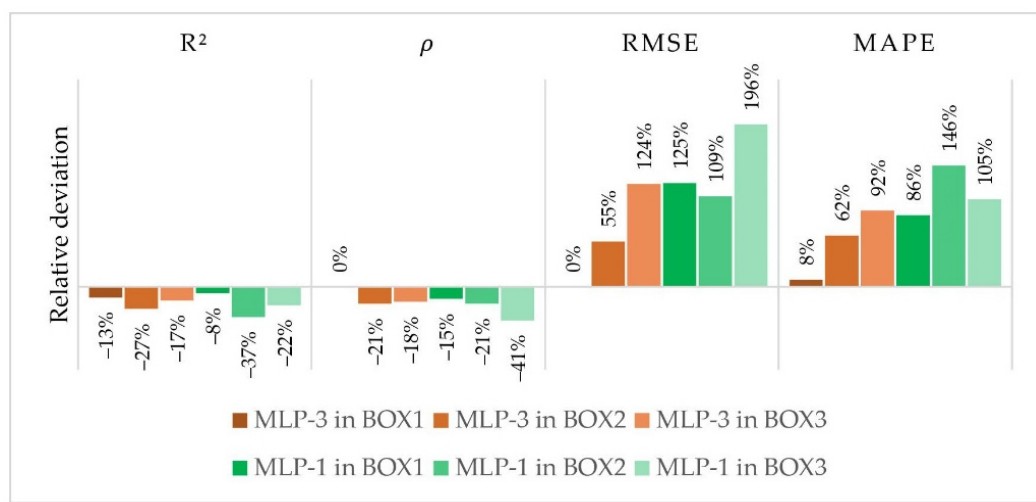

**Figure 8.** Relative deviation for MLP-3 (brown) and MLP-1 (green) models from the IDNN model in BOX1, BOX2, and BOX3.

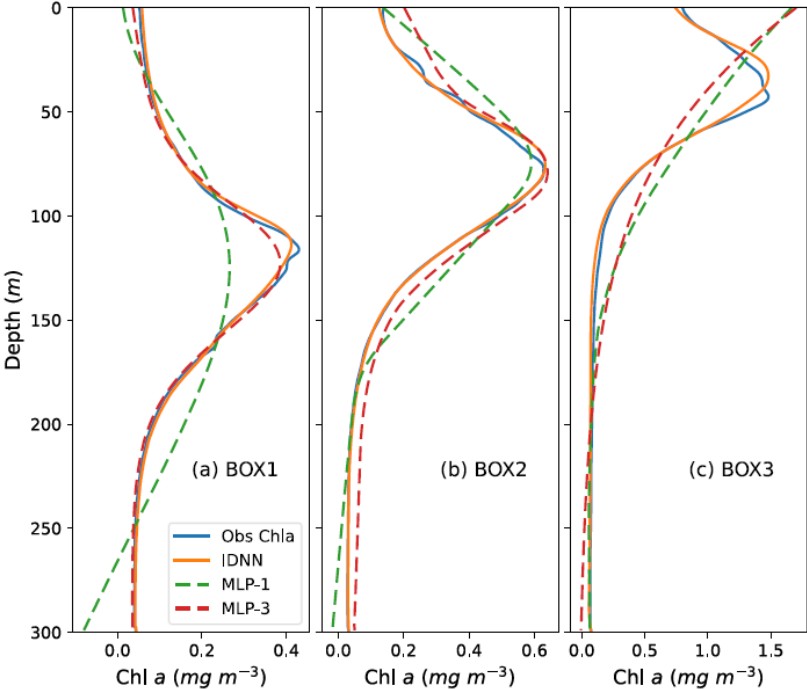

**Figure 9.** Modeling Chl *a* profiles by the IDNN and MLP models in three BOXes (**a**–**c**) of the northwestern Pacific Ocean. The configuration of MLP-1 is 6-10-1, that is, 6 input variables, 10 hidden nodes, and 1 output. The MLP-3 model is 6-3*64-1, that is, 3 hidden layers and 64 nodes in each hidden layer.

Statistically, the determination coefficients ranged within 0.47–0.79, with the average $R^2 = 0.64$ on the test set (Table 5). Analogous result was obtained by Sammartino et al. [24] by using remote-sensing data with an $R^2 = 0.63$. Here, the average value of $R^2$ was lower than that obtained by the IDNN using BGC-Argo data (the average of 0.73) (Table 3), implying a possible influence of the satellite retrieval uncertainties on the prediction accuracy, especially for the BOX3. The other statistical indexes, $\rho > 0.86$, RMSE $< 0.14$ mg m$^{-3}$ and MAPE% $< 15\%$, are almost the same as those obtained from BGC-Argo input (Table 3). The statistical results indicate that the IDNN model is robust for the remote-sensing data, with slightly less accuracy in the subpolar region.

**Table 5.** Statistical results of the comparison between the observed Chl *a* values and the predicted Chl *a* concentrations by the IDNN models using remote-sensing data. $R^2$ refers to the determination coefficient, $\rho$ to the Pearson's correlation coefficient, RMSE denotes the root mean square error, and MAPE represents the mean absolute percentage error.

| Index | Region | | |
|---|---|---|---|
| | BOX1 | BOX2 | BOX3 |
| $R^2$ | 0.79 | 0.66 | 0.47 |
| $\rho$ | 0.91 | 0.86 | 0.89 |
| RMSE | 0.0046 | 0.026 | 0.14 |
| MAPE | 0.037 | 0.076 | 0.15 |

Figure 10 shows that the mean vertical Chl *a* profile inferred by the IDNN from remote-sensing data agrees well with the mean BGC-Argo profile in each BOX, validating the good prediction accuracy of the IDNN model in the northwestern Pacific Ocean. In detail, the two mean profiles (orange and blue lines) are almost overlapped in the BOX1, except for a slightly overestimation below the SCM depth (110–140 m) (Figure 10a). The Chl *a* concentrations above the SCM depth in the BOX2 (<80 m) are relatively overestimated (Figure 10b). The scope of overestimation enlarges from the upper SCM layer to the lower part (20–80 m) in the BOX3 (Figure 10c).

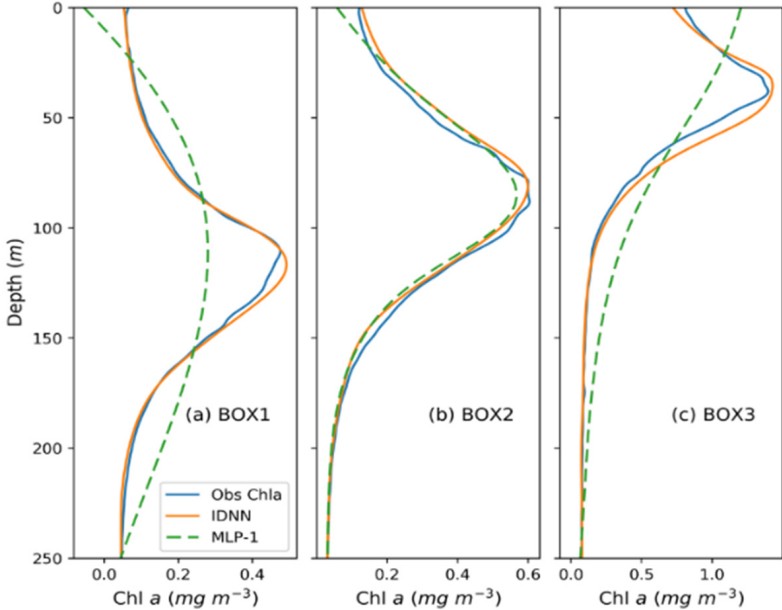

**Figure 10.** Comparisons of the mean profiles of observed values (blue line), IDNN predictions (orange line), and MLP-1 predictions (green dash line) inferred from remote-sensing data in three BOXes (**a–c**).

Moreover, the performance of IDNN is compared with the MLP-1 method by using remote-sensing data as input variables (Figure 10a–c). In the BOX1 and BOX2, the MLP-1 model captures weaker SCMs than the observed one, especially in the BOX1. While in the BOX3, no SCM phenomenon is presented in the predicted profile by the MLP-1 model. Similar results are obtained to those inferred from the BGC-Argo reported in Section 3.4. In general, the comparisons between the observed and the predicted profiles confirm the applicability of our IDNN model to extend the remote-sensing-based surface information to the subsurface layer.

## 4. Conclusions

In this study, for the first time, we developed and applied an improved DNN model with Gaussian radial basis activation function, to retrieve the vertical structure of Chl *a*

concentration and the associated SCM characteristics in the northwestern Pacific Ocean. The annually averaged SCM depth was incorporated into the bias term and the Gaussian radial basis activation function via the training process of the DNN model, which improved the prediction capability of model from surface-ocean Chl *a* data and SST. The vertical structure of Chl *a* concentration and SCM characteristics, which were estimated by our DNN model, showed a good agreement with observations in different seasons and along the trajectory of BGC-Argo floats. Compared to a series of neural network methods, our IDNN model with Gaussian radial basis activation function captured the SCM characteristics in the northwestern Pacific Ocean, especially in subpolar areas with high surface Chl *a* concentrations. Moreover, the SCM characteristics were reproduced well by our IDNN model inputting remote-sensing surface data.

This study used surface-ocean Chl *a* and SST as input variables for the IDNN model to reconstruct the non-uniform vertical Chl *a* profiles. A future improvement of our model involves employing additional input variables—such as photosynthetically active radiation, light attenuation coefficient, and oceanographic parameters (e.g., sea surface height and wind components)—that potentially affect SCM characteristics. Meanwhile, the training process of present IDNN is pixel-to-pixel without considering temporal variations of neighboring pixels, which is similar to other shallow ANN models. Thus, a deep learning technique with a combination of convolution neural network (CNN) and a long short-term memory (LSTM) neural network will be adopted to predict the target by considering the time series of the most correlated surrounding pixels.

**Author Contributions:** Conceptualization, X.G. (Xiang Gong); Data curation, J.C. and X.X.; Formal analysis, X.G. (Xinyu Guo), X.X. and K.L.; Funding acquisition, X.G. (Xun Gong), X.X. and X.G. (Xiang Gong); Investigation, J.C., X.G. (Xun Gong) and X.G. (Xiang Gong); Methodology, J.C. and X.G. (Xiang Gong); Project administration, X.G. (Xiang Gong); Resources, X.G. (Xiang Gong); Software, J.C.; Supervision, H.G.; Validation, J.C.; Visualization, J.C.; Writing—original draft, J.C.; Writing—review and editing, X.G. (Xun Gong), X.G. (Xinyu Guo) and X.G. (Xiang Gong). All authors have read and agreed to the published version of the manuscript.

**Funding:** This work was supported by the Ministry of Science and Technology of the People's Republic of China (2019YFE0125000), the National Nature Science Foundation of China-Shandong Joint Fund (U1906215), and the National Natural Science Foundation of China (41876032 and 41890805).

**Acknowledgments:** We would like to acknowledge the International Argo Program and the national programs that contribute to the BGC-Argo data, which were collected and made freely available (https://argo.ucsd.edu, https://www.ocean-ops.org, accessed on 19 April 2021). The Argo Program is part of the Global Ocean Observing System. This work was partly supported by the Ministry of Education, Culture, Sports, Science and Technology, Japan (MEXT) to a project on Joint Usage/Research Center–Leading Academia in Marine and Environment Pollution Research (LaMer). This work was also supported by the Key Laboratory of Coastal Environmental Processes and Ecological Remediation, Chinese Academy of Sciences Opening Fund (2020KFJJ04).

**Conflicts of Interest:** The authors declare no conflict of interest.

## Appendix A

**Table A1.** Basic statistical evaluations used for the assessment of the IDNN performance.

| | |
|---|---|
| Determination Coefficient | $R^2 = 1 - \frac{\sum_{i=1}^{n}(x_i - p_i)^2}{\sum_{i=1}^{n}(x_i - \bar{x})^2}$ |
| Pearson's Correlation Coefficient | $\rho = \frac{\sum_{i=1}^{n}(p_i - \bar{p})(x_i - \bar{x})}{\sqrt{\sum_{i=1}^{n}(p_i - \bar{p})^2}\sqrt{\sum_{i=1}^{n}(x_i - \bar{x})^2}}$ |
| Root Mean Square Error | $RMSE = \sqrt{\frac{1}{n}\sum_{i=1}^{n}(p_i - x_i)^2}$ |
| Mean Absolute Percentage Error | $MAPE = \frac{1}{n}\sum_{i=1}^{n}\left|\frac{p_i - x_i}{x_i}\right|$ |
| Mean Bias Error | $MBE = \frac{1}{n}\sum_{i=1}^{n}(p_i - x_i)$ |
| Mean Relative Bias Error | $MRBE = \frac{1}{n}\sum_{i=1}^{n}\frac{p_i - x_i}{x_i}$ |

Note: $n$ is the amount of data number, $p_i$ is the model estimated values and $x_i$ is the observed Chl *a*.

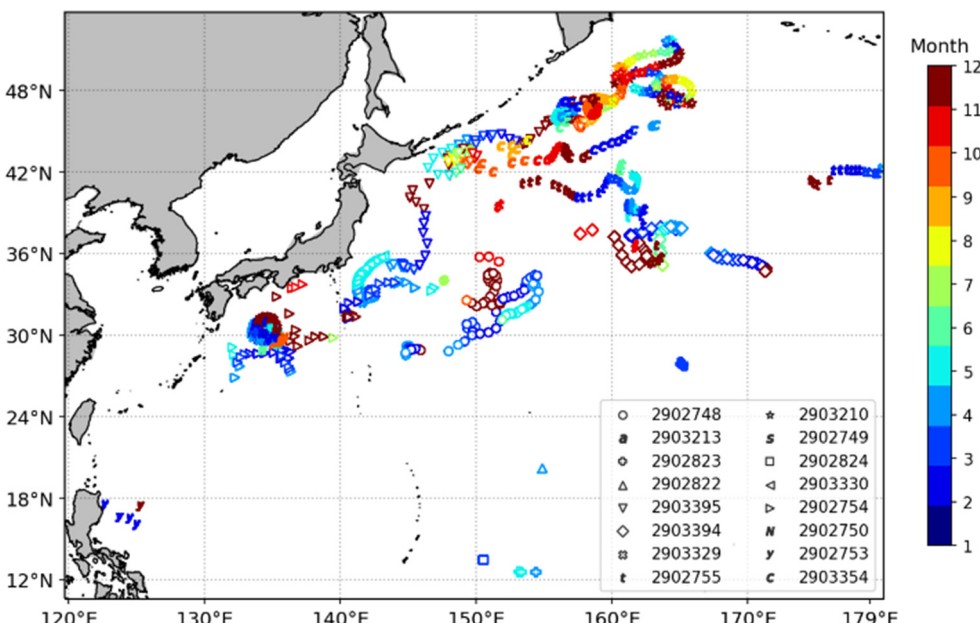

**Figure A1.** Locations and measuring months of 16 BGC—Argo profiles without SCM in the northwestern Pacific Ocean.

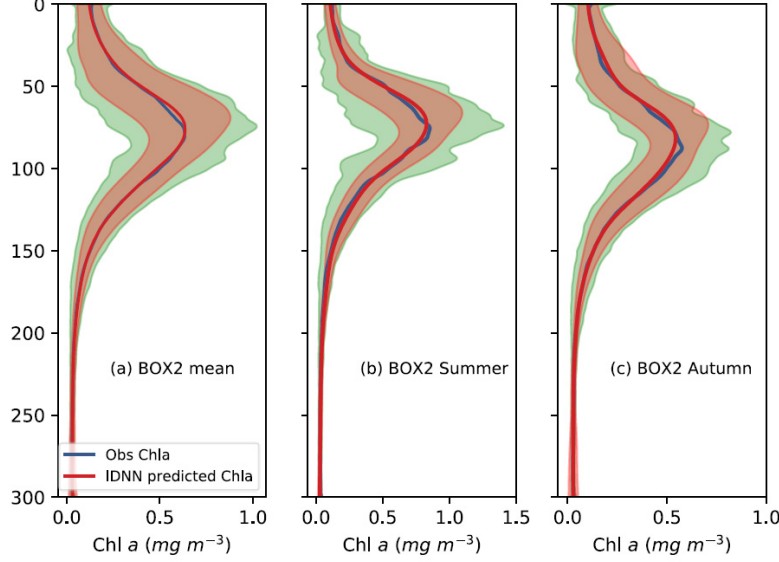

**Figure A2.** Aggregated chlorophyll vertical profiles from the test set in BOX2 during four seasons (**a**), during summer (**b**), and during autumn (**c**), after removing two profiles with extreme values (over 4 mg m$^{-3}$) between 0–50 m. The blue and red solid lines represent the mean of observed value and the mean of IDNN predicted Chl *a*, respectively. The pink and green shades are the standard variance of the model results and observations, respectively, which overlap and form the brown shade.

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
