# Peer review of "Improved Perceptron of Subsurface Chlorophyll Maxima by a Deep Neural Network: A Case Study with BGC-Argo Float Data in the Northwestern Pacific Ocean"

_remotesensing, doi:10.3390/rs14030632_

Round 1

Reviewer 1 Report

   With the advancement of BGC-Argo, retrieving the vertical profiles of Chl a with the structure of SCM from surface-ocean data has become a hot topic. Due to the complexity of SCM characteristics, many people tent to use neutral network model based on a large dataset. This study improved the performance with deep neutral network by using data collected in different areas of northwestern Pacific Ocean.

Some general and specific comments are as follows, aiming to improve the quality of this work.

  1. Lines 96-97: How to define the surface value of Chl a and temperature? Use to average value within a certain depth, or others? Please specify it.
  2. Lines 179. Variability and quality of data used for NN model is critical for determining its application. In this study, 1342 out of the total 2409 profiles were retained for training the IDNN model. As validated with the 15% profiles, the IDNN model exhibited good agreement with the observations. Even though, more descriptions are expected here for confirming the potential application of this model. How about the locations or the measuring seasons of these profiles with or without SCM? Is it possible for us to discriminate them from surface ocean data?

  1. 75% of the input data were used for training. Were they randomly selected? How about the stability by randomly selecting training data?

  1. Lines 181 -191: All of the SCM profiles were classified into 3 boxes according to the latitudes, with the intensity and depth of SCM showing obvious differences. Does it mean the weak seasonal variability of SCM characteristics? The ranges of SCM intensity and the depth are suggested to be given in Table 1.

  1. Figure 4. Since the mean value of Chl a in BOX3 is larger than that in BOX1 and BOX2. I think it’s not proper to confirm the better performances of the IDNN model in BOX1 and BOX2 than that in BOX3. We have to consider the relative variability (e.g relative bias).

  1. Figure 5. Due to the non-standard bell-shape of vertical Chl a profile, higher estimation offsets for the area with high surface Chl a value were observed, as shown in Fig.5. Since recent studies have improved the curve fitting of SCM structure by superimposing the Gaussian profile onto a linearly (Uitz et al. 2006) or exponentially decreasing background Chl concentration (Mignot A. et al. 2011), is it possible to improve the function in this study?

  1. Figure 6. It’s not easy to read this figure. Could it be replaced with scatter diagram?

Reviewer 2 Report

First of all, BGC-argo is direct observation data, not remote sensing data or satellite data. I did not find any remote sensing data in the manuscript.

Second, I cannot find the reason why readers need to read this manuscript. There is no story, no scientific significance, only DNN related tests.

Third, the abstract is not attractive, and the conclusion has no focus.

Author Response

Reply to Anonymous Referee #2

Comments and Suggestions for Authors

First of all, BGC-argo is direct observation data, not remote sensing data or satellite data. I did not find any remote sensing data in the manuscript.

Response: Thanks for this suggestion. We trained the DNN model by using surface-ocean data and BGC-Argo profiles of Chl a, which provides a feasible approach for constructing the nonlinear relationship between vertical Chl a profiles and remote-sensing data. We added the application of our DNN model in the revised version.

Second, I cannot find the reason why readers need to read this manuscript. There is no story, no scientific significance, only DNN related tests.

Response: As pointed by the reviewer, we aim to propose a new artificial neural network-based algorithm that can be used to derive vertical profiles of chlorophyll-a concentration, and subsequently the various characteristics attributed to the ubiquitous subsurface chlorophyll maximum in the global oceans.

Third, the abstract is not attractive, and the conclusion has no focus.

Response: Thanks for this suggestion. In the revised abstract and conclusion, we’ve highlighted the aim and findings of this work.

Reviewer 3 Report

Please see my detailed review attached.

Round 2

Reviewer 1 Report

I’m satisfied with the response from authors.

Author Response

Thank you very much for your time and assistance.

Reviewer 2 Report

Thanks for the author’s response,
However, the improvement of the manuscript is still limited, and it has not reached the standard that I think is acceptable. Unfortunately, I must reject this manuscript for the next step.

Author Response

Reviewer 2:

Thanks for the author’s response. However, the improvement of the manuscript is still limited, and it has not reached the standard that I think is acceptable. Unfortunately, I must reject this manuscript for the next step.

Response: We thank the reviewer's comment and make a further revision to improve the quality of this manuscript as following:

Lines 147-149, ‘Figure 2 plotted the trajectories of the 14 BGC-Argo profiling floats within 123 °E–180 °E, 12 °N–48 °N, where a SCM feature was observed. Figure A1 showed the locations of vertical Chl a profiles observed from 16 BGC-Argo floats in the absence of a SCM.’

lines 193-198, ‘The remaining 1342 profiles with SCM covered four seasons from February 2018 to April 2021 (Table 1, Figure 2), while those profiles that are excluded in the initial quality control are mostly characterized by the absence of SCM during the winter season (Figure S1). In general, vertical Chl a profiles with a SCM accompany higher SSTs and lower surface Chl a concentrations in subtropical and subpolar areas, compared to the profiles without SCM.’

Lines 215-225: ‘The average value of SCM depth and its intensity in each BGC-Argo float is shown in Table 1. The averaged SCM intensity from profiles of each BGC-Argo float in tropical area (12 °N–24 °N) has ranges between 0.41–0.67 mg m-3 over depths of 115–153 m; the averaged SCM intensity in subtropical (26 °N–38 °N) is about 0.65–1.2 mg m-3 with shallower SCM depths over 40–89 m. At high latitudes (38 °N–48 °N), the SCM exists at a depth <50 m with the largest intensity larger than 1.5 mg m-3. Table 1 also presents the seasonal averaged SCM depth and intensity. Compared with summer and autumn, the SCMs get deeper and weaker in winter and spring, which is probably due to increased vertical mixing in winter and spring. This indicates that physical entrainment may extract some of phytoplankton from the SCM layer to the surface layer and thereby reduce the SCM intensity [2, 32].’

lines 324-328: ‘In Figure 5f-g, large standard variances of observed profiles occurred between 0-50 m (green shades), which is similar to those in Figure 4e. Therefore, we removed the two profiles with an episodically stronger SCM within 0-50 m in BOX2 during summer and autumn and then the two variance spikes in 0-50 m disappear (Figure A2).’

lines 336-339: ‘The high standard variance in surface Chl a (e.g. the green shaded areas in Figure 5h-k) is related to dissimilar shapes of vertical Chl a profiles, and thus reduce the network prediction accuracy on SCM characteristics.’

lines 358-369: ‘The SCM thickness can be estimated by fitting vertical Chl a profiles using the generalized Gaussian function (Equation 3). Since recent studies have developed the curve fitting of vertical Chl a profiles by superimposing the generalized Gaussian function onto a linearly [19] or exponentially decreasing background Chl a concentration [33], we compared the curve-fitting performance of the three approaches in 3 BOXes. The results show that the generalized Gaussian function has a higher goodness of fit than other two functions. For example, for the non-standard bell-shape of vertical Chl a profiles in BOX3 (e.g. Figure 5i-k), the generalized Gaussian function has the goodness of fit of 90%, following by the function combined an exponentially decreasing background (88%) and the function combined a linearly decreasing background (about 76%). Based on our statistical results, we estimate the SCM thickness by using the generalized Gaussian function (Equation 3).’

lines 486-488: ‘Constructing the nonlinear relationship between vertical Chl a profiles and remote-sensing data is the next phase of this study, especially retrieving SCM characteristics pixel-to-pixel, based on the advantage of a combination between a convolution neural network and a long short-term memory neural network in considering the target and neighborhood pixels.’

Reviewer 3 Report

Please see PDF attached.
